# A SiCN Thin Film Thermistor Based on DVB Modified Polymer-Derived Ceramics

**DOI:** 10.3390/mi13091463

**Published:** 2022-09-03

**Authors:** Chao Wu, Fan Lin, Xiaochuan Pan, Yingjun Zeng, Guochun Chen, Lida Xu, Yingping He, Daoheng Sun, Zhenyin Hai

**Affiliations:** Department of Mechanical and Electrical Engineering, School of Aerospace Engineering, Xiamen University, Xiamen 361005, China

**Keywords:** polymer-derived ceramics, thermistor, SiCN, conductivity

## Abstract

Carbon-rich SiCN ceramics were prepared by divinylbenzene (DVB)-modified polysilazane (PSN2), and a high-conductivity SiCN thin film sensor suitable for medium-low temperature sensing was fabricated. The modified liquid precursors were patterned by direct ink writing to produce SiCN resistive grids with line widths of several hundreds of micrometers and thicknesses of several micrometers. The introduction of DVB not only increases the critical thickness of SiCN ceramics several times, but also significantly improves the conductivity of SiCN, making it meet the conductivity requirements of sensing applications in the mid-low temperature range. The electrical conductivity and microstructure of DVB-modified SiCN ceramics were studied in detail. In the temperature range of 30~400 °C, the temperature resistance performance of DVB modified SiCN resistance grid was measured. The SiCN ceramics with low DVB content not only have excellent electrical conductivity, but also have good oxidation resistance.

## 1. Introduction

Polymer-derived ceramics (PDCs), such as SiCN, SiOC and SiBCN, are considered for high-temperature sensor applications due to their semiconducting behavior and excellent thermal stability [1,2,3,4,5]. The unique polymer-to-ceramic (liquid-solid) transition process allows for attractive patterning options, such as direct ink writing (DIW) and soft lithography, making PDCs a promising candidate for high-temperature micro-electromechanical systems (MEMS) and thin-film sensors (TFSs) candidate material [6,7,8]. Temperature sensors are a promising application of PDCs as thermistors. A thermistor is a device that measures temperature by measuring its resistance [9]. Manganate, SiCN, SiOC and other thermistors with negative temperature coefficient (NTC) show exponential reduction in resistance with temperature, while positive temperature coefficient (PTC) thermistors such as Pt and NiCr show a linear increase in resistance with temperature [10,11,12]. The in-situ generated free carbon phase plays a key role in the electrical conductivity and semiconducting behavior of PDCs [13,14,15,16], which enables the electrical resistance of PDCs to have a pronounced temperature response.

Although the temperature-resistance behavior of some PDCs has been developed, it is not easy to fabricate TFSs using PDC materials. The poor electrical conductivity of PDCs and the low critical thickness of constrained sintered PDCs films remain huge obstacles for the wider application of PDC TFSs. For example, poor electrical conductivity makes PDCs electrically insulating at room temperature, which limits their operating temperature range to above 400 °C [17,18]. The critical thickness of constrained sintering is about 3 μm, which limits the thin-film of PDCs device [1]. Although the relatively good conductivity can be obtained by molecular design of precursor solutions and optimization of pyrolysis parameters, the pyrolysis temperature is usually higher than 1400 °C [13]. Another effective strategy is to fill the PDC with conductive particles that exceed the penetration threshold [6,7]. This strategy not only improves the electrical conductivity of PDCs film but also improves their critical thickness. However, the dispersibility and stability of solid-phase particles in the precursor solution are difficult to guarantee. Therefore, liquid-phase modified PDCs would be preferred. Divinylbenzene (DVB), as a widely used precursor of carbon sources, has been used to improve the electrical conductivity of PDCs [19,20,21]. For example, the conductivity of DVB-modified PDC-SiCN is improved by 5 orders of magnitude [11]. However, currently, DVB-modified PDC devices are usually millimeter-scale and difficult to integrate with structural components, which significantly decreased the sensitivity and response time of the sensor [11]. The advantages of PDCs in additive manufacturing and controlled molding have not been well utilized.

In this study, the electrical conductivity of PDC-SiCN was increased by adding a DVB carbon source precursor to a commercially available polysilazane (PSN). An in-situ integrated thin-film temperature sensor was successfully fabricated via DIW platform based on the Weissenberg effect and tested from room temperature (30 °C) to 400 °C, expanding the application of PDC sensors in the mid-low temperature range.

## 2. Materials and Methods

### 2.1. Materials and Fabrication Methods

In this study, a commercially available polysilazane (PSN, Institute of Chemistry, Chinese Academy of Sciences, Beijing, China) was utilized as the SiCN precursor, DVB (technical grade 80%, Sigma-Aldrich, St. Louis, MO, USA) was selected as the carbon source to improve the conductivity of SiCN. The fabricated process of TFSs was illustrated in Figure 1a. First, different mass fractions of DVB were added to PSN and magnetically stirred at 100 °C for about a few minutes to form a uniform printable mixture (The contents of DVB were 10 wt%, 20 wt%, 30 wt%, 40 wt%, 50 wt%, respectively). Briefly, DVB-modified PSN2 ink was printed by a Weissenberg-based DIW platform, which consisted of three key components: an x–y high-precision moving platform, a homemade printing setup including a printing head, and a charged–coupled device camera. Then, the printed thin-film resistor grids were pyrolyzed in a tube furnace under high-purity nitrogen atmosphere at 1100 °C for 4 h.

### 2.2. Characterization Techniques

Resistance grid thicknesses were determined by a profilometer (Dektak XT). SEM (SUPRA55 SAPPHIRE) coupled with EDS was used to characterize the morphology and elemental content of the obtained films. XPS (Thermo Scientific ESCALAB Xi+) measurements were performed to determine chemical bonds. Free carbon in SiCN was characterized using confocal in situ Raman spectroscopy (LabRAM HR Evolution). As shown in Figure 1b, the utilised temperature–resistance.

## 3. Results

### 3.1. Film Morphology

Pores and cracks are the main factors affecting PDCs conductivity [1]. Therefore, the critical thickness of the DVB-modified SiCN film (defined as the maximum thickness of monolayer deposition) was first determined. In the DIW process, the line width of the resistance grids was fixed at about 500 μm, and the line thickness was successively increased. After pyrolysis, the thickness of the resistance grids and the shape of the surface profile were determined by a profilometer. As shown in Figure 2, the critical thickness of SiCN film gradually increased from 3.7 μm of SiCN to 5.7 μm of 10 wt% DVB-SICN, 6.0 μm of 30 wt% DVB-SiCN, 6.9 μm of 50 wt% DVB-SiCN. The addition of DVB provides more carbon sources for SiCN ceramics. The high critical thickness may be related to the carbon content of SiCN. The increased critical thickness reduces the risk of film cracking and peeling during DIW patterning and pyrolysis, which is beneficial for maintaining the structural integrity of the resistive grids.

Figure 3a shows an optical image of the thin film sensor. Figure 3b is a SEM image showing a SiCN resistive grid line, indicating that its line width is about 550 μm. Figure 3c shows the surface morphologies of DVB20-SiCN film. It can be seen that the film is dense without obvious defects. The SEM image of the cross-section of the SiCN film shown in Figure 3d shows that the SiCN film is well bonded to the Al_2_O_3_ substrate. The results of elemental analysis of all films are summarized in Table 1, and it can be clearly seen that the addition of DVB significantly increases the carbon content in the SiCN films. The higher oxygen content may be related to the oxygen contamination during the DIW process, and the oxygen adsorption on the surface of the SiCN film.

### 3.2. Film Composition

The XPS was employed to analyze the chemical composition and bonding characteristics within the film. The XPS spectra of DVB10-SiCN film pyrolyzed at 1100 °C are shown in Figure 4. The spectrum of Si (2p) shows peaks at 102.35 eV and 104.19 eV due to the formation of Si-N, Si-O bonds. For the C (1 s) spectrum, peaks were observed at 284.8 eV, 286.06 eV, and 288.25 eV and are attributed to C-C/C-H, C-O/C-N, and C=O, respectively. The highest intensity of the C-C peak indicates the completion of the pyrolysis process and the formation of free carbon. Similarly, for N (1 s) spectrum, the peaks at 399.35 eV, and 400.95 eV correspond to C-N and Si-N, respectively. As for the O 1 s spectrum, two peaks centered at 532.4 eV and 533.87 eV, respectively, are associated with the Si-O band and adsorbed -OH, which is related to the oxygen contamination during the preparation [1]. The above results indicate the formation of free carbon-rich SiCN ceramics.

Raman spectroscopy was used to further analyze the carbon in the SiCN films. The Raman spectra of the investigated SiCN ceramic films are shown in Figure 5. The spectra of SiCN modified with different DVB concentrations exhibit similar shape, which contains a D peak at ~1333 cm^−1^, and G peak at ~1610 cm^−1^, indicating a strong disorder state of the amorphous carbon. The intensity, position, and width of the D and G bands may vary, depending on the structural organization of the sample under study [22]. The intensity ratio of the D and G modes, *I_D_*/*I_G_*, enables the evaluation of the carbon nanoparticle size by using the Ferrari-Robertson equation [22,23]:(1)$$\frac{I_{D}}{I_{G}}=C^{\prime }\left(\lambda \right)L_{a}^{2}$$
C′(λ) is a coefficient depending on the excitation wavelength of the laser. The value of C′(λ) for the wavelength of 532 nm is assigned to 0.6195 nm^−2^ [24]. The *L_a_* values are listed in Table 2. The lateral cluster size La in all samples is between 1 nm and 2 nm, indicating the nanostructural nature of the free carbon in SiCN. Previous studies have shown that carbon in PDC SiCN will undergo precipitation during the pyrolysis process, and then evolve into free carbon [24]. The value of *I_D_*/*I_G_* is used to measure the disorder degree of free carbon, and the higher the ratio is, the higher the defect carbon content would be [25]. When the DVB content is 20 wt%, the *I_D_*/*I_G_* value suddenly increases, implying that the content of disordered carbon in SiCN gradually increases. The increase in *L_a_* may be due to the in-plane growth of nano-polycrystalline graphite [23].

### 3.3. Electrical Performance

The sensing properties of the thin film resistance grids were characterized by measuring the temperature-dependent resistance. The SiCN resistance grid pyrolyzed at 1100 °C is insulated at room temperature (conductivity: 10^−6^ S/m–10^−4^ S/m) [5,26]. The electrical conductivity of the DVB-modified SiCN films was calculated from the resistance and size of the resistive grid. The room temperature conductivities of the DVB10-SiCN, DVB20-SiCN, DVB30-SiCN, DVB40-SiCN and DVB50-SiCN resistor grids are 9.4 S/m, 39.8 S/m, 32.3 S/m, 24.8 S/m and 30 S/m, respectively (Figure 6a). Compared with SiCN, the electrical conductivity of DVB-modified SiCN is significantly improved. However, when the DVB content was increased to 20%, the conductivity did not further improve. This is related to the content of free carbon in SiCN and the percolative network formed by it. In conductive composites, when the conductive phase reaches the percolation threshold, the conductivity increases by orders of magnitude. Further increasing the concentration of the conductive phase, the conductivity increases slowly and becomes stable [27]. The percolation behavior can be attributed to the formation of free carbon network. In DVB20/SiCN the percolative network is already completely formed and it does not improve significantly in DVB50/SiCN [28]. This explains that the electrical conductivity of the two compositions is quite similar despite their DVB amounts of 20 wt% and 50 wt%, respectively.

The temperature sensing properties of thin-film resistive grids were characterized by measuring temperature-dependent resistance (Figure 6b–l). Below 350 °C, the resistance of the SiCN resistive grid is greater than 1 GΩ, which limits the application of SiCN TFSs in the mid-low temperature range. The high conductivity of DVB-modified SiCN resistive grid makes it more suitable for sensing in the mid-low temperature range. In the temperature range of 30–400 °C, all DVB-modified SiCN resistive grids exhibit negative temperature coefficient of resistance, that is, the resistance decreases monotonically with increasing temperature, presenting a good sensitivity of DVB-modified SiCN resistive grids under high temperature (T) environments. The ln(1/R)–1000/T curves for the DVB-modified SiCN resistive grids were obtained according to the thermistor equation [17,18]:(2)$$ln\frac{1}{R}=c_{1}\frac{1}{T}-c_{2}$$
where *c*_1_ and *c*_2_ are constants. For all SiCN resistive grids, linear behavior following the thermistor equation was obtained. The constant *c*_1_ values of DVB10-SiCN, DVB20-SiCN, DVB30-SiCN, DVB40-SiCN, and DVB50-SiCN thermistors are −372.35, −509.72, −478.65, −388.51 and −811.69, respectively, indicating that they all have excellent sensitivity to temperature change. However, during one cycle of heating and cooling tests, it can be found that SiCN films modified with low DVB content (DVB: 10–30 wt%) have better repeatability. The incomplete coincidence of the heating and cooling resistance curves in the temperature range of 200–400 °C is related to the response speed of the thermocouple and the thermal conduction behavior of thermal components such as Al_2_O_3_ substrates. In other words, the thermocouple measures the ambient temperature of the Al_2_O_3_ substrate, while the temperature experienced by the TFS is closer to the surface temperature of the Al_2_O_3_. During the heating and cooling process, their thermal conduction behaviors are different, resulting in incompletely consistent resistance curves. However, when the heating and cooling are completed, the resistance of the TFS can return to its original value. For the SiCN films modified with high DVB content (DVB: 40–50 wt%), it can be found that the resistance of TFS increases significantly after one thermal cycle. This may be related to the oxidation of carbon. The modified SiCN films with low DVB content not only significantly improved the electrical conductivity, but also exhibited better resistance repeatability.

## 4. Conclusions

In this study, a DVB-modified PDC thin-film resistive grid with a line width of about 550 μm and a thickness of less than 7 μm was fabricated for by a DIW process. Compared with PDC-SiCN, the introduction of more carbon sources by adding DVB not only significantly improves the electrical conductivity of SiCN, but also increases the critical thickness of SiCN film several times. This provides another method for the thinning, patterning and high conductivity of PDC devices. A thin thermistor integrated with the Al_2_O_3_ substrate was successfully fabricated using DVB-modified PDC-SiCN as the sensing element, which demonstrated the feasibility of the proposed method. The conductive behavior and microstructure of DVB-modified PDC-SiCN ceramics were studied in detail. The temperature resistance behavior of the DVB-modified SiCN resistive grid in the mid-low temperature range from room temperature to 400 °C was measured. The SiCN ceramics modified with low DVB content not only effectively improved the electrical conductivity but also had better oxidation resistance.

## Figures and Tables

**Figure 1 micromachines-13-01463-f001:**
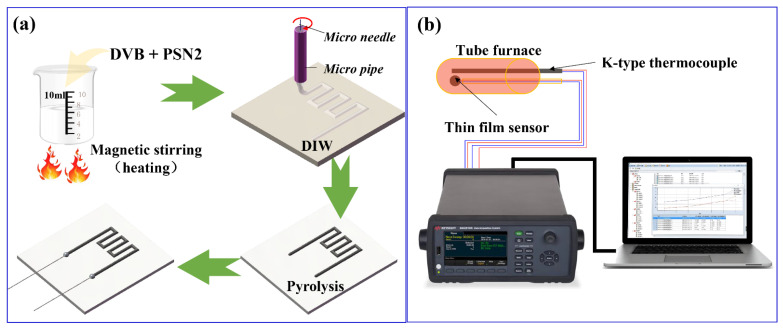
Schematic illustration of (**a**) process and (**b**) temperature-resistance test device.

**Figure 2 micromachines-13-01463-f002:**
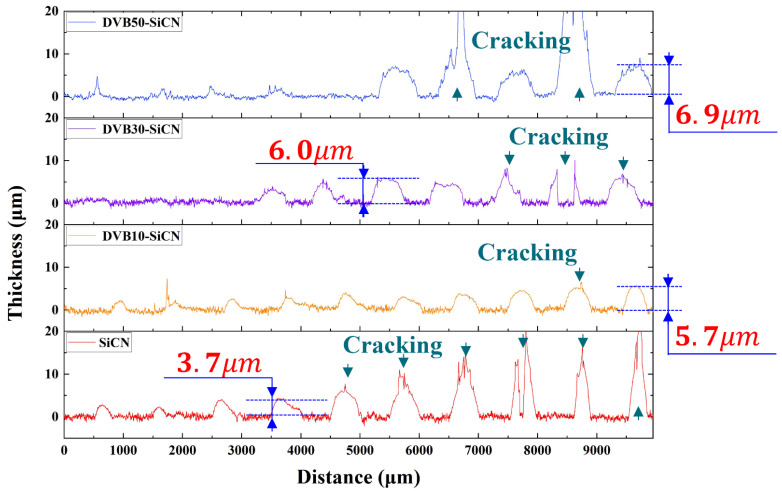
Surface profile of SiCN and DVB-modified SiCN lines.

**Figure 3 micromachines-13-01463-f003:**
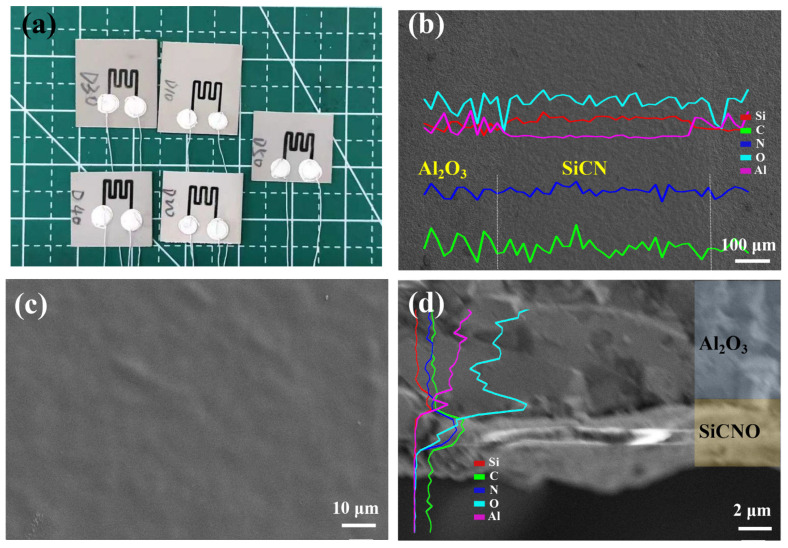
(**a**) Optical image showing SiCN thin film sensor. (**b**) SEM image showing a single DIW-SiCN line (inset shows EDS analysis conducted in the line scan mode). (**c**) Surface morphology of the DVB20-SiCN film. (**d**) SEM image showing the cross section of the DVB20-SiCN film (inset shows EDS analysis conducted in the line scan mode).

**Figure 4 micromachines-13-01463-f004:**
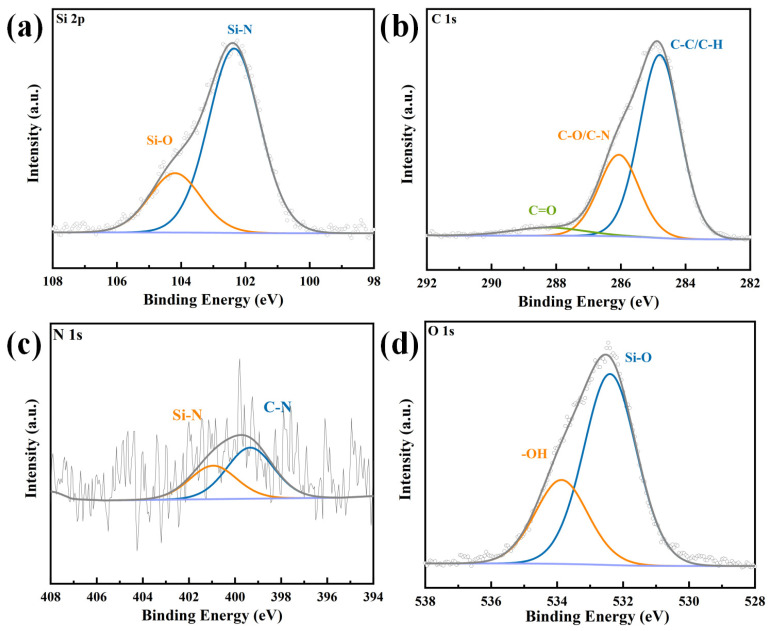
XPS spectra of the as-prepared DVB20-SiCN film. (**a**) Si 2P spectrum. (**b**) C 1s spectrum. (**c**) N 1s spectrum. (**d**) O 1s spectrum.

**Figure 5 micromachines-13-01463-f005:**
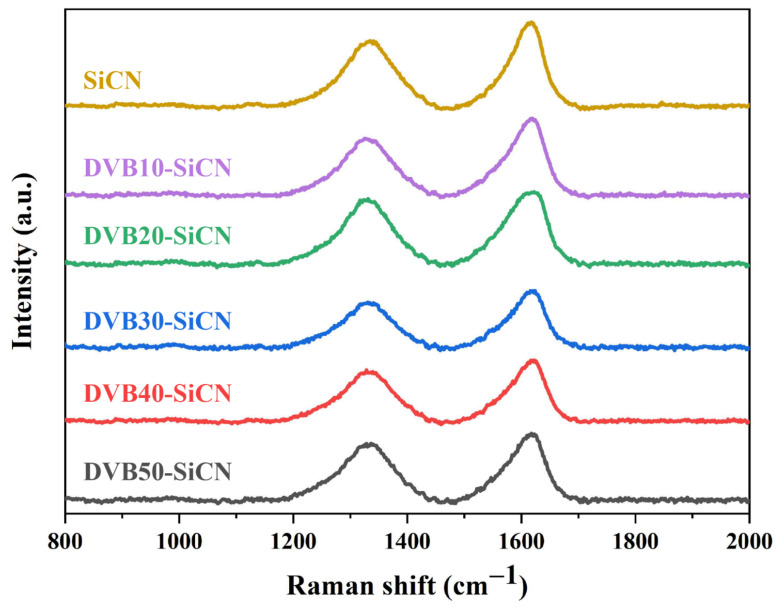
Raman spectra of SiCN films modified with different concentrations of DVB.

**Figure 6 micromachines-13-01463-f006:**
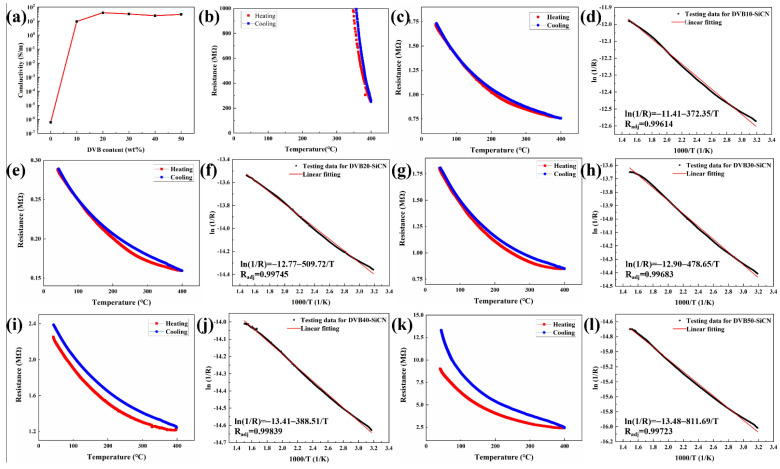
(**a**) Variation of SiCN conductivity at room temperature with DVB content. The resistance of film as a function of temperature: (**b**) SiCN film, (**c**) DVB10-SiCN film, (**e**) DVB20-SiCN film, (**g**) DVB30-SiCN film, (**i**) DVB40-SiCN film, (**k**) DVB50-SiCN film. A plot of resistance vs. temperature in the format ln(1/R) vs. 1000/T: (**d**) DVB10-SiCN film, (**f**) DVB20-SiCN film, (**h**) DVB30-SiCN film, (**j**) DVB40-SiCN film, (**l**) DVB50-SiCN film.

**Table 1 micromachines-13-01463-t001:** Results of elemental analysis.

Sample	Si (wt%)	C (wt%)	N (wt%)	O (wt%)
SiCN	32.75	17.89	2.02	47.34
D10-SiCN	31.87	19.24	3.23	45.67
D20-SiCN	32.64	18.99	4.84	43.53
D30-SiCN	33.35	20.01	2.31	44.33
D40-SiCN	29.46	23.29	5.52	41.72
D50-SiCN	31.20	22.69	3.92	42.19

**Table 2 micromachines-13-01463-t002:** The intensity ratio of D peak to G peak *I_D_*/*I_G_* and carbon cluster size *L_a_* obtained from the curve-fitting of Raman spectra.

Sample	SiCN	DVB10-SiCN	DVB20-SiCN	DVB30-SiCN	DVB40-SiCN	DVB50-SiCN
***I_D_*/*I_G_***	0.77	0.76	0.89	0.80	0.84	0.87
***L_a_* (nm)**	1.118	1.106	1.198	1.133	1.630	1.183

## Data Availability

The data used to support the findings of this study are included within the article.

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
