# Peer review of "A SiCN Thin Film Thermistor Based on DVB Modified Polymer-Derived Ceramics"

_micromachines, 2022, doi:10.3390/mi13091463_

Round 1

Reviewer 1 Report

The authors should include the following in their revised manuscript

1) Literature about thermistors derived from other materials.

2) Should distinguish between positive coefficient and negative coefficient thermistor action

3) Revise or remove figures 3d, 3e and 3f. Ther is no clarity in them

4) A therortical modelling at atomic level

Reviewer 2 Report

In this manuscript, the authors prepare a carbon-rich SiCN ceramic by modifying polysilazane (PSN2) with divinylbenzene (DVN), which effectively enhances the conductivity of SiCN and prepares a high conductivity SiCN thin film sensor suitable for low and medium temperature sensing. This study effectively improved the electrical conductivity of ceramics with SiCN, but the manuscript still needs to be improved in some details. I hold the opinion that this manuscript can be accepted in Micromachines after minor revisions. The details are as follows

1. Please explain the sudden increase in the ID/IG ratio of sample DVB20-SiCN in Table 2.

2. Please adjust the scale of the vertical coordinate of Figure 5a to facilitate comparison with the rest of the figures.

3. To facilitate the interpretation of the percolation thresholds in "3.3 electrical performance", it is recommended that the conductivity values of the different samples be plotted and analyzed.

Reviewer 3 Report

Manuscript ID: micromachines-1886976

Title: SiCN Thin Film Thermistor Based on DVB Modified Polymer-Derived Ceramics

This paper reports a DVB modified SiCN thin film thermistor fabricated by direct ink writing and high temperature pyrolysis. The conductivity of the film was modified by addition of DVB making it suitable for sensing applications in the mid-low temperature range. The work is interesting. However, the current manuscript needs some revisions before possible acceptance for publication.

Comments and suggestions:

1. Did the authors fit the relation between the resistance of the thermistor and temperature? An accurate relation between the two is required for application as a thermistor.

2. How did the authors measure the elemental fraction of the ceramics? Please describe the method in ‘Materials and Methods’ part. Besides, the precursors are PSN and DVB, why the obtained ceramics contain very large amount of oxygen (more than 40 wt% as shown in Table 1)?

3. Page 5, Figure 3. There seems no obvious difference in the elemental distribution along the line scanning between the Al2O3 and SiCNO part in figure 3(b). How did the author distinguish the two parts from the SEM and EDS results? The EDS results in figure 3(c) seems didnt match the morphology either. Besides, the SiCNO in figure 3(b) is supposed to be SiCN?

4. Usually, the machines information should be removed and replaced with standard scale markers in micrographs (figure 3). Please check out the requirement of this journal and revise if needed.

5. Page 6, last line. The author claim The carbon size increases with increasing DVB concentration. However, the results in Table 2 did not reveal such conclusion. The La of DVB20-SiCN is largest. Please check out carefully. ‘The lateral cluster size La in all samples is between 1 nm and 2 nm, indicating the nanostructural nature of the ceramics.’ This is not accurate. It can only reveal the nanostructure of the free carbon instead of the ceramics.

6. Page 8. What can be deduced from figure 5 is that the repeatability during heating and cooling was very poor for the SiCN film modified with high DVB content. However, we can not claim that “For the SiCN film modified with high DVB content (DVB: 40wt%-50wt%), it can be found that the TFS dergoes significant oxidation after one thermal cycle” just based the resistance-temperature plot. Of course, the oxidation is the reason for the poor repeatability. But you can not draw the conclusion on the oxidation issue directly.

7. There are many typos in the manuscript. For example: Page 3, line 90, DVB-SICN should be DVB-SiCN. page 5, line 114, DVB-SiCN should be DVB50-SiCN. page 8, line 163/164, DVB20/SiCN is supposed to be ‘DVB20-SiCN’. Careful revision throughout the manuscript is mandatory required.

Round 2

Reviewer 1 Report

The authors incorporated all the suggestions given in the revised manuscript

The revised manuscript can be accepted for publication

Reviewer 3 Report

The comments have been well addressed in the revised manuscript. I recommend 'accept' for publication.